# Childhood-Onset ANCA-Associated Vasculitis: From Genetic Studies to Advances in Pathogenesis, Classification and Novel Therapeutic Approaches

**DOI:** 10.3390/ijms252413704

**Published:** 2024-12-22

**Authors:** Liching Yeo, Asma Naheed, Chantelle Richards, Coziana Ciurtin

**Affiliations:** 1Department of Paediatrics, Homerton University Hospital, London E9 6SR, UK; liching.yeo@nhs.net; 2Department of Paediatric Rheumatology, Great Ormond Street Hospital, London WC1N 3JH, UK; asma.naheed@gosh.nhs.uk; 3Department of Paediatrics, Southend University Hospital, Westcliff-on-Sea SS0 0RY, UK; chantelle.richards@nhs.net; 4Department of Adolescent and Young Adult Rheumatology, University College London NHS Foundation Trust, London NW1 2PG, UK; 5Centre for Adolescent Rheumatology, Division of Medicine, University College London, Rayne Building, London WC1E 6JF, UK

**Keywords:** antineutrophil cytoplasmic antibodies (ANCA)-associated vasculitis, children-onset, genes, classification, management

## Abstract

Childhood-onset antineutrophil cytoplasmic antibodies (ANCA)-associated vasculitis (AAV) represents a heterogeneous group of multi-system autoimmune conditions associated with chronic inflammation, characteristically affecting small blood vessels, leading to various organ and system manifestations. Although rare in paediatric populations, AAV poses challenges in early recognition, diagnosis and management of refractory cases. This review highlights the characteristics of clinical presentation and outcomes of AAV in children, as well as its current classification and progress achieved in understanding the disease pathogenesis, with a focus on adult and paediatric genetic studies. Furthermore, we discuss the management of AAV in children, as well as new emerging therapies and future research needs, while proposing a potential algorithm for a childhood-onset-AAV therapeutic approach based on the disease phenotype.

## 1. Introduction

Antineutrophil cytoplasmic antibody (ANCA)-associated vasculitis (AAV) constitutes a group of rare diseases characterised by necrotising inflammation of small blood vessels and the presence of ANCA [1], usually directed against myeloperoxidase (MPO) or proteinase-3 (PR3) [2,3].

AAV is a severe systemic autoimmune disease affecting the small blood vessels, particularly in the lungs, kidneys and skin [2,3]. The condition is better characterised in adults, and has three major clinicopathological variants of AAV, including granulomatosis with polyangiitis (GPA), microscopic polyangiitis (MPA) and eosinophilic granulomatosis with polyangiitis (EGPA). A key feature of AAV is the presence of ANCA.

Paediatric AAV is extremely rare in childhood. Data on this entity is scarce, and the condition is less well characterised than in adults. The annual incidence rate for AAV was reported by a Swedish study as 3.2 (95% CI, 1.1–5.4) per million children, with an estimated incidence of 1.4 (95% CI, 0–2.8) per million children for both GPA and MPA and much lower for EGPA, 0.4 per million children [4]. The estimated prevalence is 3.41–4.28 (95% CI, 2.33–6.19) per million children [5].

Increasing rates of paediatric AAV [6,7], similar to those reported in adult studies [8], have been recently observed. This is likely due to increased clinical recognition and efforts to improve the disease classification across the life span. In contrast to adult AAV, paediatric AAV has a higher female preponderance. The median age of diagnosis has been found to be between 10.7 and 14 years, with a median duration of symptoms prior to diagnosis of approximately 2 years. Similar to adult populations [9], the most frequent type of childhood AAV is GPA, followed by MPA and EGPA [4].

Despite the progress achieved in providing harmonised recommendations for the standard of care in childhood vasculitides [10], there are still unmet needs for a better understanding of the pathogenic particularities of the childhood phenotypes, which can support better research, aiming for improved outcomes for children and young people with AAV.

In this review, we aim to provide an update on the evidence available in relation to clinical presentation, serological markers, and classification and diagnosis of childhood AAV. Additionally, we will explore the emerging evidence provided by genetic/epigenetic studies in childhood AAV and advances in understanding their contribution to the disease pathogenesis, which can support future therapeutic progress in AAV with childhood onset. Although more studies are available in adult AAV compared to children, we compared and contrasted findings derived from these studies, rather than extrapolating data from adults to children, while providing additional suggestions for the management of childhood AAV based on phenotype and pathogenesis similarities while discussing the unmet needs for future research, and providing suggestions for addressing them.

## 2. Clinical Presentation of Childhood AAV

AAV is a multi-system condition comprising three distinct clinico-pathological phenotypes which also have overlapping features. Similar to other systemic vasculitides, non-specific constitutional symptoms of fever, lethargy and weight loss are common, occurring in 80% of AAV paediatric cases [11].

There are a few notable ongoing initiatives in place aiming to underpin large cohort studies supported by collaborative agreements and well-defined data collection protocols, such as the European Alliance of Associations for Rheumatology/Paediatric Rheumatology European Society (EULAR/PReS) vasculitis working group involved in several multi-centre AAV studies [12], the US/Canada Childhood Arthritis and Rheumatology Research Alliance (CARRA) AAV working group [13], the Paediatric Rheumatology International Trials Organisation (PRINTO) vasculitis cohort study and database [14], in addition to expert initiatives, such as the 2008 Ankara Consensus Conference [15], which all have been instrumental in advancing the knowledge related to childhood AVV.

### 2.1. GPA [14]

GPA is a systemic vasculitis with granulomatous inflammation of small and medium-sized vessels that usually manifests as a triad of symptoms, frequently but not exclusively affecting the upper and lower respiratory tracts and the kidneys. Upper respiratory tract involvement tends to be the most frequent manifestation, followed by constitutional symptoms, renal, lower respiratory tract, musculoskeletal and cutaneous involvement [16]. GPA in children tends to be more severe than in adults, with early aggressive subglottic stenosis and/or nasal septum perforation being reported in the literature [17]. Other ear, nose and throat (ENT)-associated symptoms frequently encountered include epistasis, sinusitis, mastoiditis, serous otitis media and hearing loss. The lower respiratory tract symptoms usually include chronic cough, breathlessness and haemoptysis. Renal involvement at the onset was reported in 65% of children [16], occurring over a spectrum that ranges from isolated urinary sediment abnormalities to rapidly progressive glomerulonephritis.

### 2.2. MPA

MPA is a vasculitis that predominantly affects small vessels without granulomatous involvement. In children, MPA tends to present with constitutional symptoms or a pulmonary-renal syndrome, and often, there is a delay in diagnosis [18]. Pulmonary involvement is commonly associated with chronic cough, chest pain, haemoptysis or anaemia. Over 40% of children had associated pulmonary haemorrhage [11], while extensive glomerular involvement usually manifests as focal segmental glomerulonephritis associated with proteinuria, hypertension and haematuria is also common [6]. There is an increased risk of progression to end-stage renal disease within 5 years from diagnosis in children and young people presenting with pulmonary-renal involvement [19].

MPA is also considered part of a clinical spectrum, as its clinical features overlap with GPA manifestations, while both subtypes have similar non-pulmonary histologic findings and are associated with the presence of ANCA. While most MPA cases test positive for anti-MPO-ANCA with a perinuclear pattern, while GPA cases more frequently demonstrate anti-PR3 ANCA staining with characteristic cytoplasmic distribution (see below), both types of ANCA could be found in either condition.

### 2.3. EGPA

EGPA is a rare presentation in children, manifesting as a predominantly small-vessel vasculitis, with a prolonged prodromal course of asthma and hypereosinophilia leading to pulmonary infiltrates and accumulation of extravascular eosinophils resulting in systemic vasculitis [20].

### 2.4. AAV-Associated Glomerulonephritis in Children

One of the key features of AAV is the presence of renal involvement, more commonly associated with necrotising crescentic glomerulonephritis (NCGN). GN is associated with both GPA and MPA in a higher proportion in children compared to adults across various studies (65–78% vs. 50–65% in GPA and 88–95% vs. 70–85% in MPA, respectively), and overall there was no association between ANCA positivity or ANCA titre and renal remission in children, while in adults, a higher relapse rate was seen in PR3 vs. MPO—ANCA-positive cases [21].

The ARChiVE/Pediatric Vasculitis Initiative (PedVas) registry analysis, including 406 children with AAV (76.4% with renal involvement) followed up for more than 15 years, has been published recently [22]. The registry evaluated the predictive value of the ANCA serotype on renal outcomes and found a higher proportion of girls with renal disease and more cases with MPO-ANCA positivity compared to PR3-ANCA with associated renal involvement (n = 111, 88.1%, vs. n = 179, 77.2%). Additionally, more MPO-ANCA-positive cases had impaired kidney function at baseline, estimated by a glomerular filtration rate (eGFR) below 50 mL/min/1.73 m^2^ compared to the PR3-ANCA-positive group (35.7% vs. 14.2%).

In another study including 85 children with AAV-GN, the majority classified as crescentic GN (50.6%), followed by focal (21.2%), sclerotic (15.3%), and mixed (12.9%) AAV-GN found that eGFR at baseline differed significantly in relation to histopathology features, with the crescentic and sclerotic classes having a lower baseline eGFR compared to the focal and mixed classes [23]. A comparative analysis with adult AAV-GN showed a female predominance of renal involvement in children vs. male in adults, differences in histology (with a higher proportion of crescentic lesions found in children and more focal and mixed GN classes in adults), and greater renal plasticity in children, with improved recovery and better outcomes compared to adults [21].

## 3. Outcomes in Childhood AAV

A recent retrospective multi-centre French study found that severe renal, ENT, and pulmonary involvement remain the main causes of significant morbidity in young people despite advances in treatment and wider access to B cell depletion therapies [24]. In this cohort, P-ANCA positivity was associated with relapse in 81% of cases, and 54.8% of cases already chronic kidney disease at 1 year post-diagnosis. A large retrospective multi-centre study including 337 children with ANCA-associated glomerulonephritis from 41 different countries found that 40% of cases required kidney replacement therapy after a median duration of 26 months post-diagnosis and that this was predicted by an increased peak of creatinine during the first 3 months after presentation [25]. Pulmonary involvement is another major cause of significant morbidity, with 1/3 of children presenting with diffuse alveolar haemorrhage or massive haemoptysis, while 10% had respiratory failure in a North-American cohort of 105 children with AAV [26], while MPA was the most common cause for severe pulmonary involvement in another single-centre cohort study [27]. Despite improvement with treatment, only 42% of children with AAV achieved remission, and 63% had a present at 12 months follow-up [26], suggesting an unmet need for improved outcomes.

The overall mortality rate associated with childhood AAV was found to vary from 3.6% in a French retrospective cohort [24] to 5% in a global cohort of ANCA-associated glomerulonephritis [25].

## 4. Serological Markers Associated with AAV

### 4.1. Characteristic ANCA

The presence of circulating ANCA autoantibodies supports the diagnosis of AAV but is not universal and does not necessarily differentiate between AAV phenotypes, although vasculitic manifestations are more commonly observed in ANCA-positive cases with EGPA, while eosinophilic manifestations are more common in ANCA-negative individuals [28,29,30,31,32,33]. Two main localisation patterns have been described on indirect immunofluorescence: one cytoplasmic (c-ANCA) and one perinuclear (p-ANCA), and their usual target antigens are neutrophil granule proteins, PR3 and MPO, respectively. While paediatric AAV data are limited, Cabral et al. reported in 231 total cases (48 MPA, 183 GPA) PR3-ANCA and/or c-ANCA positivity in 67% GPA and 17% MPA in young people with AAV, and MPO-ANCA and/or p-ANCA positivity in 55% MPA and 26% GPA cases [11]. EGPA and limited disease phenotypes (particularly involving the upper respiratory airways) are often ANCA-negative. In a small cohort of 33 children/young people with EGPA, 25% were ANCA-positive, typically p-ANCA/MPO-ANCA [34].

The exact underlying mechanisms and pathogenic role of ANCA have not been completely elucidated; it is also recognised that they involve innate immune cell priming leading to the translocation of MPO/PR3 to the plasma membrane, facilitating ANCA binding on the cell surface, resulting in subsequent activation of neutrophils and monocytes, conducive to blood vessel wall adhesion, inflammation and endothelial injury, and ultimately localised necrosis and immune cell infiltration which perpetuates inflammation [35]. The direct pathogenic effect of ANCA antibodies in driving disease pathogenesis has been demonstrated in numerous animal studies [36,37].

### 4.2. Other ANCA Associated with AAV

Aside from PR3 and MPO, ANCAs can target other molecules derived from neutrophils, such as moesin, defensin, elastase, LAMP-2 (lysosome-associated membrane protein-2, expressed on the glomerular endothelium). In general, these “minor” ANCAs are considered to have low pathogenicity [38], although anti-moesin has been associated with the development of small-vessel vasculitis and has been detected in most MPO-positive AAV cases by reacting with neutrophils and monocytes to induce inflammatory cytokines and chemokines [39].

LAMP-2 has also been detected in adults with AAV-associated renal disease, with moderate to high LAMP-2 titres also found in 35% of paediatric systemic vasculitis cases involving small-medium vessels, with the highest concentrations in MPO- or PR3-positive individuals [40]. Some PR3-positive cases also had antibodies reacting to complementary-PR3 (cPR3), encoded by the antisense RNA of the PR3 gene [41]; plasminogen has been identified as its target, implicating it as an autoantigen in PR3-ANCA vasculitis [42].

Pentraxin-3 is a novel ANCA antigen that may play a role in AAV pathogenesis. Anti-pentraxin-3 antibodies were found in nearly 40% of cases with MPA, GPA or EGPA; approximately 35% of both MPO and PR3 ANCA-positive individuals and half of AAV cases without MPO or PR3 ANCA at diagnosis [43].

Anti-lactoferrin antibodies are characteristic of a subgroup of patients with EGPA, characterised by a higher frequency of renal involvement and increased serum C-reactive protein levels and disease activity. Lactoferrin is usually released from neutrophil-specific granules after their activation and acts as an endogenous suppressor of NET formation. Subsequently, anti-lactoferrin antibodies promote NET formation triggered by neutrophil activation [44].

### 4.3. Other Antibodies with Clinical Relevance in Paediatric AAV

ANCA is associated to a certain extent with anti-glomerular membrane (GBM) antibodies, being found in 20–40% of children with anti-GBM disease, while 5% AAV are positive for anti-GBM antibodies [45]. The double-positive children experienced more severe renal involvement, requiring stronger immunosuppression and plasma exchange therapy. The presence of antinuclear antibodies (ANA), found in 50% of cases, and double-stranded DNA antibodies (dsDNA), found in 20.6% of cases in a large cohort of 218 children with AAV (older than 14 years), was associated with poorer renal outcomes in a large Chinese study [46], which prompted the authors to strongly recommend renal biopsy to ensure the optimal management of these cases.

## 5. Classification of Childhood AAV

Classification criteria for AAV were developed to facilitate research by defining consistent and homogenous disease populations to investigate in outcome studies clinical trials of various interventions, as well as defining clinical protocols. Classification criteria have not been validated for clinical diagnosis but are frequently used in clinical practice to guide diagnosis. Failure to meet the classification criteria does not preclude a specific diagnosis, as the classification criteria will not support the diagnosis of atypical cases.

The most widely used classification system has been the American College of Rheumatology (ACR) 1990 [47], derived from adult data and endorsed by the Chapell Hill Consensus in 1994 [48]. Over time, limitations of the ACR criteria in adult populations have been identified [49]. Because of differences in clinical presentation according to the age at disease onset, the criteria performed poorly within paediatric populations [13,47].

In 2005, EULAR/PReS proposed a consensus-based framework for classifying paediatric vasculitis [50]. This was revised by the EULAR/PRINTO/PReS in 2008 in Ankara [15]. Although these criteria were adapted from the adult ACR criteria, they are more commonly used in childhood AAV, as they were derived from paediatric vasculitis cohorts. The framework of the EULAR/PRINTO/PReS classification is based on the size of the involved blood vessels.

The second Chapel Hill Consensus was held in 2012 and provided a revised framework for the classification of vasculitis and specific definitions. Alongside this consensus, an initiative named ‘Diagnostic and classification criteria in vasculitis study’ was founded to collect adult vasculitis data to support further refinement of the adult classification criteria, which led to the 2022 ACR/EULAR criteria for the classification of AAV [51], which provided the first classification for MPA. A recent study evaluated the performance of the EULAR/PRINTO/PReS Ankara 2008 and 2022 ACR/EULAR classification criteria for GPA in children and found adequate performance (sensitivity of 94.8% and 89.6%, and specificity of 95.3% and 96.3%, respectively) [12].

## 6. Pathogenesis of Childhood-Onset AAV: From Genetic Factors to Dysregulation of the Immune Pathways

While the pathogenesis of AAV remains unclear, various factors such as genetic susceptibility, environmental factors and altered innate and adaptive immune responses are all considered contributing factors. In fact, 20% of AAV risk is estimated to be due to genetic factors [52]. Much of the knowledge about paediatric AAV and its management has been adapted from adult studies [13,53]. Gibson et al. suggested that given the early onset of childhood AAV (12–14 years) [11] vs. adult-onset (>50 years) [54], a greater influence of genetics over environmental factors could be hypothesised [55]. Additionally, they also highlighted other observations in paediatric AAV compared to adult AAV, such as female preponderance in paediatric cases (opposite to the male predominance in adult AAV) [6,11], low prevalence of mild disease, more frequent constitutional upset, and renal and pulmonary involvement in children compared to adults [11,55,56,57].

## 7. Genetic Associations with AAV

Despite large-scale studies conducted on AAV to investigate genetic susceptibilities, only a few significant loci have been identified, and not all have been validated across various adult populations, which is very likely because of disease heterogeneity, as well as its low prevalence. Recent research has highlighted a genetic overlap across various types of vasculitis in children and also unveiled new genetic loci associated with AAV at the genome-wide significance level [58], which has potential implications for drug-repurposing strategies.

Genome-wide association studies (GWAS) and candidate-gene association studies have investigated single nucleotide polymorphisms (SNPs) to identify disease-susceptibility loci associated with AAV [59]. The second approach has been used more extensively in adult AAV studies and has identified several genes within the human leucocyte antigen (HLA) region [60,61], as well as genes in the non-HLA region encoding ANCA-associated proteins, such as *SERPINA1* [62], *PRTN3* [63], or immunoregulatory genes for AAV, such as *PTPN22* [64], *FCGR3B* [65,66] and *TLR9* [67]. Some of these findings have been confirmed by GWAS analyses, which have additionally identified other AAV susceptibility loci [68,69,70,71].

There is a scarcity of Mendelian randomisation studies in adult AAV [72], while these are completely lacking in childhood AAV, and therefore, establishing the causative role of the genetic associations described by various studies is still a challenge. However, many of the gene variants identified have implications in biological processes relevant to AAV pathogenesis as well as in the selection of potential therapeutic targets, which are highlighted below.

### 7.1. GWAS in GPA/MPA

Since 2012, several GWAS analyses in adult AAV have been conducted—two for MPA and GPA [68,70], one for GPA [69] and one for EGPA [71], all including adult populations of European ancestry. A recent GWAS study of 2687 Northern European Caucasian individuals with GPA and MPA suggested that GPA is associated with genetic variants of *HLA-DP*, *SERPINA1* (encoding α1-antitrypsin), and *PRTN3* (encoding PR3), whereas MPA is mainly associated with *HLA-DQ* [73].

The first GWAS of paediatric AAV, including 63 paediatric AAV patients (42 GPA, 15 unclassified, 6 MPA) and 315 adult controls of European ancestry, was conducted to analyse genetic susceptibility factors previously found in adult AAV, such as *HLA-DP* and *HLA-DQ* [55]. Analysis revealed significant association only with *HLA-DPB1*04:01* (*p* = 1.5 × 10^−8^, odds ratio [OR] = 3.5) and a stronger association with PR3-ANCA than MPO-ANCA

None of the non-HLA loci associated with adult AAV (*SERPINA1*, *PRTN3*, *SEMA6A*, and *PTPN22*) reached genome-wide significance in the paediatric AAV cohort. However, a significant association with AAV diagnosis was demonstrated with two regions on chromosomes 2 and 5, corresponding to *ankyrin repeat* and *SOCS* (suppressor of cytokine signalling) *box* containing *ASB3* (*p* = 7.9 × 10^−6^, OR 2.9), and *LINC02147* (*p* = 5.7 × 10^−6^, OR 5.7) genes, respectively, which was unique to the paediatric AAV cohort [55].

Several AAV genetic associations are relevant for providing insight into disease pathogenesis and potential therapeutic approaches.

*ASB3 (Ankyrin Repeat And SOCS Box Containing 3)* is a negative regulator of TNF receptor 2 (TNFR2)-mediated cell response to tumour-necrosis factor alpha (TNF-α) [74], shown to have a proinflammatory role in lupus nephritis and ANCA-associated glomerulonephritis, and the effectiveness of anti-TNF-α medications is being assessed in clinical trials [75].

*LINC02147 (long intergenic non-protein coding RNA 2147)* is an RNA gene of the long non-coding RNA class implicated in inflammation, fibrosis and tumourigenesis [76].

Another non-HLA region of interest was found on chromosome 20, mapped to *ADNP (activity-dependent neuroprotector homeobox*) (ADNP; rs2870024) (*p* = 3.0 × 10^−7^, OR 6.6), which is a transcription factor encoding NAP neuroprotective peptide that downregulates proinflammatory cytokines (TNF, IL-16, and IL-12) with relevance for AAV pathogenesis [77].

Further research is required to understand the significance of these findings and their potential mechanistic implications in paediatric AAV pathogenesis.

Gibson et al. found that paediatric and combined paediatric and adult AAV cohorts had enhanced T cell receptor and interferon signalling pathways; this is consistent with whole blood gene expression profiling of paediatric and adult AAV patients, showing differential gene signatures relevant to T cell receptor and interferon signalling with therapeutic implications [78].

### 7.2. Candidate-Gene Association Studies in GPA/MPA

#### 7.2.1. HLA Region

Human leukocyte antigens located on chromosome 6p21 are genes with pivotal roles in disease and immune defence, especially antigen presentation. Candidate-gene association studies have been undertaken in various populations (Swedish, Germans, Italians in Europe, and Japanese and Chinese in Asia). AAV susceptibility has been found with *HLA-DPA1*, *HLA-DPB1*, *HLA-DQA1*, *HLA-DQB1* and *HLA-DRB1* [79]. In GPA, consistency across populations was only found with *HLA-DPB1*0401* in German cohorts [60,80]. GWAS analyses have also demonstrated strong associations with AAV and polymorphisms in *HLA*-*DPB1* [68,69,70], particularly with PR3-ANCA positivity compared to GPA [59]. MPO-ANCA positivity has also been found to be associated with *HLA*-*DQA* and *DQB1* loci in GPA and MPA [68,70], *DRB1*0901* [81,82], *DRB1*1302* [83], *DQA1*0302* [84], and *DQB1*0303* [81] in MPA in East Asian cohorts, and *HLA-DQ* in EGPA in Europeans [71]. This highlights the genetic distinctions and strong association between genetic background and ANCA specificity rather than the clinical phenotype/AAV subtype.

Notably, *HLA* alleles have shown associations with disease severity, prognosis and relapse—*HLA-DRB1**04:05 with poor renal prognosis, *HLA-DRB1**04:02 with high mortality [85] and *DPB1**04:01 with risk of relapse [86].

The most notable genetic correlations were found with the *Retinoid X Receptor Beta (RXRB) gene* found in a GPA cohort of Northern German patients and with [60] *theRING finger protein 1 gene*, which encodes one of the E3 ubiquitin-protein ligases, found to be strongly associated with GPA in ANCA-positive subjects [80].

#### 7.2.2. Non-HLA Region

Several genetic risk factors for AAV have been identified with non-MHC coding genes, such as PTPN22, CTLA-4, SERPINA1, PRTN3, CD226, TLR9, IRF5, NOTCH4, AGER, and CFB [87].


*Proteinase 3 (PRTN3)*


*PRTN3* is located on 19p13.3 and encodes for the expression of PR3, which is the main antigen for ANCA, leading to reduced neutrophil activation and endothelial adhesion relevant for AAV pathogenesis [88].


*Serpin Family A Member 1, SERPINA1*


The *SERPINA1* gene is located on 14q32.13, and encodes α1-antitrypsin (α1-AT), an inhibitor of serine proteases, including PR3 [88]. GPA has been found to be associated with *SERPINA1* variants, leading to consequent PR3 accumulation [62,89] and blood vessel damage [59].


*Protein Tyrosine Phosphatase, Non-Receptor Type 22 (PTPN22)*


*PTPN22* is located on 1p13.2 and encodes lymphoid tyrosine phosphatase (Lyp), which negatively regulates T cell activation [90]. A meta-analysis of four studies in 1399 white patients with ANCA disease and 9934 normal controls found that *PTPN22* 620W was associated with AAV [91].


*Cytotoxic T Lymphocyte-Associated Antigen-4 (CTLA4)*


*CTLA4* is located on 2q33.2 and encodes a glycoprotein expressed on activated T cells, which inhibits T cell activation and immune response by binding to CD80/CD86 on antigen-presenting cells. Conversely, CD28, which competes with CTLA4 for CD80/CD86 binding, exerts a stimulatory signal. Abatacept binds to the costimulatory molecules CD80 and CD86 on antigen-presenting cells to block CD28 interaction with T cells. An open-label prospective trial of abatacept involving 20 patients (aged 17–73 years) with non-severe relapsing GPA found it to be successful and well tolerated—90% had disease improvement, 80% achieved remission at a median time of 1.9 months [92].

### 7.3. Genetic Associations with EGPA

Genetic studies on EGPA are limited due to its rarity in adults and, to a greater extent, paediatric populations.

#### 7.3.1. GWAS in EGPA

Lyons et al. demonstrated that EGPA is a polygenic disease and identified 11 loci associated with EGPA in a cohort of 676 European adult cases: *BCL2L11*, *TSLP*, *HLA-DQ*, 10p14, *CDK6*, *IRF1/IL5*, *BACH2*, *LPP*, *GPA33*, *HLA* and 12q21 [71]. Many of the genetic variants associated with EGPA were also associated with eosinophil count in the general population, implying that a tendency to eosinophilia underpins EGPA susceptibility in the context of necessary genetic or environmental triggers.


HLA


The HLA locus was associated with the MPO-ANCA-positive subset of EGPA but not the ANCA-negative group, and the strongest disease risk was found with *HLA-DRB1*08:01*, *HLA-DQA1*02:01* and *HLA-DRB1*01:03*, whereas protection was conferred by *HLA-DQA1*05:01* [71].


Other Gene Variants


*GPA33* (glycoprotein A33) encodes a cell surface glycoprotein that is important for barrier function in the intestinal epithelium [93] and potentially in bronchial tissue [94].

*TSLP*, *IRF1*/*IL5*, *GATA3*, *LPP*/*BCL6*, *CDK6*, *BIM* and *BACH2* encode proteins involved in TH2 responses and eosinophilic inflammation, both key in EGPA, and many of these genes have also been implicated in asthma [59].

#### 7.3.2. Candidate-Gene Association Studies in EGPA


HLA Region


Several small cohort candidate-gene studies of Italian and German EGPA patients indicate genetic association with *HLA* alleles—*HLA-DRB4* (increases the likelihood of vasculitic manifestations), DRB1*07, DRB3 (reduced frequency in EGPA versus controls) and DRB1*13 (also underrepresented in the EGPA cohort) [61,95].

Fc gamma receptors (FcγRs) bind to IgG targets, such as immune complexes and opsonised pathogens, and have effects on various processes, including phagocytosis, antigen presentation, and apoptosis of natural killer and T cells. Fcγ receptor 3B (FCGR3B) deficiency (low copy number CN ≤ 1) is associated with systemic autoimmune diseases [96] and EGPA risk, in particular vasculitic manifestations [66].

Functional variants of the pleiotropic cytokine IL-10 gene were found to be associated with ANCA-negative EGPA, but not GPA [97].

Table 1 highlights the genetic polymorphisms associated with childhood AAV.

Several new mutations leading to an AAV-like clinical presentation have been described in children. Interestingly, some mutations led to disease onset within early childhood, while other mutations were associated with adolescent and early adulthood disease onset and diagnosis (see Table 2).

Although in the majority of cases, childhood AAV is a polygenic disease, as described above, several novel mutations have been associated with AAV in children. In Table 3, we describe several characteristics that could raise suspicion of a genetic cause for an AAV-phenotype presentation in childhood. Early recognition of these mutations is relevant for therapeutic management, disease prognosis, and family planning.

## 8. Environmental Factors Associated with AAV

Although no specific studies have been conducted in childhood AAV, a recent systematic review including AAV cases older than 16 years at diagnosis highlighted the potential role of pollution in the pathogenesis of AAV, with silica and environmental factors associated with natural disasters and farming, UV-radiation and various infections (with the most convincing association being that with *Staphylococcus aureus*) being the main candidates [106].

### Epigenetics

Recent epigenetic studies have uncovered dysregulation of DNA methylation and histone modification, influencing the transcriptional regulation of ANCA-target genes with implications in AAV pathogenesis [107,108,109].

## 9. Dysregulation of Immune Pathways Involved in AAV Pathogenesis

Various immune cells have been implicated in AAV pathogenesis. There are recognised shared features between the pathogenesis of MPA, GPA, EGPA and even drug-induced AAV, with MPA being considered the prototypic AAV [35]. Although data about the disease pathogenesis are derived mainly from animal models and adult studies, childhood AAV seems to be part of the same spectrum as adult disease.

The origin of the ANCA autoimmune response is unknown but appears to involve genetically determined HLA specificities influencing autoimmune response development by initiating an immune response to a peptide that is complementary to the autoantigen, leading autoantibody responses, or through the mechanism of autoantigen and microbe-derived molecular mimicry resulting in the generation of antibodies that cross-react with the autoantigen [110].

Neutrophils play a central role in their dual capacity as effector cells responsible for endothelial damage and targets of autoimmunity. Under certain conditions, resting neutrophils in the bloodstream undergo a process known as “priming”, whereby they display target antigens (e.g. MPO, or PR3) on the surface of the membranes. Priming may be caused by a number of processes, including treatment-related reactions, infections, and activation of the alternative complement pathway [111].

Due to their ability to generate neutrophil extracellular traps (NETs), neutrophils support the presentation of ANCA autoantigens by breaking the immune tolerance to specific self-antigens (such as MPO and PR3), which are in turn presented by dendritic cells to CD4+ T cells [112,113], leading to further differentiation of B cells into autoantibody (ANCA) producing plasma cells via IL-21 production. Additionally, B cell-activating factor (BAFF) produced by activated neutrophils leads to B cell activation and differentiation [113].

Excessive neutrophil activation within small vessels is controlled by semaphorin 4D (CD100), and CD100 upregulation leads to pronounced NET-mediated vessel injury, which is also reflected in increased serum levels of active AAV [35].

Additionally, neutrophils activated by ANCA also perpetuate the activation of complement C3 and its cleavage into C3a and C3b, further leading to the generation of the powerful neutrophil chemoattractant C5a and the membrane attack complex C5b-9. This complement activation amplifies neutrophil influx, neutrophil activation, and vessel damage, resulting in aggressive necrotising inflammation associated with AAV [110].

In GPA, the characteristic granulomatous formation is hypothesised to be related to *Staphylococcus aureus* infection, which activates tissue-resident cells which release proinflammatory cytokines, leading to the recruitment of monocytes and neutrophils, and subsequent release of reactive oxygen species and lytic enzymes as antibacterial mechanisms, leading to focal necrosis. Additionally, the differentiation of recruited monocytes into macrophages that secrete IL-23 results in Th17 phenotype differentiation and increased local IL-17 production, which is critical for granuloma formation [35].

EGPA is characterised by prominent infiltration of eosinophils, through a Th2 cytokine-mediated mechanism involving elevated IL-4, IL-5 and IL-13 levels and eosinophilia. The C-C motif chemokine 26 (CCL26; also known as eotaxin 3) released from vascular endothelial cells leads to eosinophilic infiltration of the vascular wall, followed by secretion of eosinophilic granules, including eosinophilic neurotoxin, major basic proteins and eosinophilic cationic proteins, leading to tissue destruction [35].

### Pathogenesis of AAV-Related Glomerulonephritis

Histologically, ANCA-associated GN is characterised by pauci-immune immunofluorescence, necrotising and crescentic lesions in the affected glomeruli on light microscopy, and subendothelial oedema, micro-thrombosis and degranulation of neutrophils on electron microscopy. In some cases, focal deposition of immunoglobulins and complement fractions, especially C3, have been detected, and they have been associated with poorer prognosis [114].

GN in MPA differs from GPA-associated GN, and it is characterised by a lack of granulomatous lesions and frequent features of chronicity (e.g., glomerulosclerosis, fibrous crescents, interstitial fibrosis). In addition to glomerular involvement, renal disease in AAV is also associated with tubulointerstitial nephritis, and tubular lesions are important predictors of outcome, especially in patients treated with B cell depletion therapy [115].

A few novel biomarkers of renal activity associated with AAV-GN have recently been identified, such as plasma colony-stimulating factor 1 receptor (CSF1R), which correlates positively with disease activity, initial serum creatinine, and C1Q levels, but negatively with the glomerular filtration rate (GFR) in adults with AAV-GN [115]. In animal studies, the administration of sub-nephritogenic anti-GBM antibodies enhanced the reactivity of anti-MPO antibodies against rat MPO, leading to glomerular crescent formation, neutrophil infiltration and development of NETs, as well as increased glomerular expression and serum expression of TNF-α, CXCL1, CXCL2 and CXCL8, and therefore recapitulating the human GN phenotype [116].

## 10. Diagnosis of Childhood AAV

The diagnosis of AAV in children is based on expert opinion, guided by the presence of relevant clinical and histopathological features, as well as serum autoantibody testing to assess for the presence of ANCA and other autoantibodies to exclude other conditions. Diagnostic confirmation with laboratory tests, with or without histological confirmation, is recommended before treatment is initiated. ANCA positivity is commonly found in GPA and MPA [24,26], although it has low specificity for AAV in children [117]. Both the 2012 revised International Chapel Hill Consensus Conference Nomenclature of Vasculitides and the EULAR/PRINTO/PRES classification criteria for GPA in childhood are commonly used to guide diagnosis [15,118]. Recent research showed adequate performance of the 2022 EULAR/ACR adult classification criteria in paediatric AAV [12], and they are likely to be used more in the future to support diagnosis in clinical practice.

Obtaining a tissue biopsy from the kidney, nasopharynx, sinuses or lungs and immunohistochemistry (as well as light and electron microscopy for renal biopsies) is recommended to support the diagnosis of AAV [10], but it should not delay treatment initiation, especially in cases with severe organ or life-threatening manifestations. Various imaging investigations tailored to clinical manifestations, such as computer tomography (CT) or magnetic resonance imaging (MRI), could be helpful for diagnosis. Novel imaging modalities, such as positron emission tomography (PET) combined with MRI to minimise radiation, can be helpful in selected cases.

## 11. Management of Paediatric AAV

The European initiative Single Hub and Access Point for Paediatric Rheumatology in Europe (SHARE) guidelines provide general recommendations for the treatment of AAV in children [10], while CARRA has recently published consensus treatment plans which will be tested in a large observational study [119]. Despite the lack of evidence for the efficacy of therapeutic interventions for childhood AAV, a large survey indicated clinicians’ preference for childhood-specific treatment guidelines rather than using adaptations of adult AAV recommendations [120]. However, many recent adult guidelines integrate paediatric-specific recommendations, such as the ACR/Vasculitis Foundation guidelines [121] and the recent Kidney Disease: Improving Global Outcomes (KDIGO) 2024 clinical practice guidelines [122].

Glucocorticoids are the first line of therapy, as they can promptly bring inflammation under control. The recommended starting dose for severe AAV, according to the SHARE guidelines, is 1–3 pulses of 10–30 mg/kg of intravenous methylprednisolone (max 1 g), followed by 1–2 mg/kg day of oral prednisone (max 60 mg) decreased to 0.8 mg/kg/day after one month and by 0.1/0.2 mg/kg/day monthly down to 0.2 mg/kg or 10 mg daily (whichever is lower) at 6 months. The evidence of non-inferiority of a reduced-dose regimen of glucocorticoids compared to a standard-dose regimen in the adult PEXIVAS study [123], which otherwise did not show any added benefit of plasmapheresis, suggests that a faster tapering regimen could also be tried in children with AAV. Although the SHARE guidelines recommend three to four weekly intravenous cyclophosphamide as induction treatment, with rituximab and mycophenolate being reserved for selected cases, there is evidence for increased use of rituximab following its approval for use in children older than 2 years of age. Rituximab is now considered the first-line induction and maintenance therapy for children with severe AAV, following the success of the phase IIb global study, which tested its efficacy in 25 children with GPA and MPA [124]. It is recommended at a dose of 375 mg/m^2^ body surface area intravenously weekly for 4 weeks, in addition to glucocorticoids.

The KDIGO recommendations recognise that there is a place for combining cyclophosphamide with rituximab as induction therapy for severe cases with rapid renal function deterioration [122], although caution is required due to the high toxicity risk.

Despite the results of the PEXIVAS study, which showed no benefit from plasmapheresis and included a small number of children older than 15 years, it has been argued that plasmapheresis can still be used in severe cases, with rapid deterioration, especially in the context of lack of response to other therapies.

Additional prophylaxis treatment for *Pneumocystis jirovecii* pneumonia (PJP) during treatment with cyclophosphamide, rituximab, and/or high-dose glucocorticoids, as well as gonadal protection with triptorelin or sperm preservation in the context of treatment with cyclophosphamide, should be part of shared decision-making with young people and their families to minimise the risk of immunosuppression-associated toxicity.

The most commonly used and recommended maintenance treatment options are azathioprine, methotrexate, mycophenolate, or rituximab [10]. Although the ACR recommendations for the management of AAV in adults favour remission maintenance with rituximab over azathioprine or methotrexate, with the last two being preferred over mycophenolate and leflunomide [121], a real-life retrospective study showed that mycophenolate was the most commonly used maintenance therapy in children with AAV [24]. Long-term maintenance treatment with rituximab should not be guided by ANCA titres or by the dynamics of B cell repopulation, as they do not correlate well with flares; longer scheduled rituximab maintenance treatment (46 months) was associated with increased flare-free survival compared to 18 months of rituximab (96% vs. 74%) [125]; therefore, the Canadian recommendations for adult AAV suggest a minimum of 2 years for maintenance treatment with rituximab, extended to an additional 18 months for high-risk cases [126].

Extracorporeal membrane oxygenation (ECMO) may be required as adjuvant therapy for severe or refractory respiratory failure in the context of diffuse alveolar haemorrhage. A registry study identified a survival rate of 76% in 34 children with AAV who received ECMO, with poorer prognosis found in those who developed renal failure while on ECMO, and higher mortality compared to adults in the context of diffuse alveolar haemorrhage (37% vs. 17%) [127]. ECMO is therefore reserved for use in severe cases, potentially minimising the risk of ventilation-related lung injury.

Treatment with intravenous immunoglobulins (IVIG) can also be explored in refractory cases, based on adult recommendations [128].

## 12. New Emerging Therapies

Although we recognise that the newly licensed treatment options for AAV have been evaluated in adult cohorts, there is currently enough evidence related to shared pathogenesis, adequate performance of classification criteria, and unified established treatment approaches for AAV across the life span to support our proposal for the potential use of novel therapeutic approaches for childhood-onset AAV, as detailed in Figure 1. We also discuss the limitations of extrapolating therapeutic strategies from adult to childhood-onset AAV.

Newly licensed treatments have already been integrated with the therapeutic armamentarium for adult AAV, such as avacopan (targeting the complement component C5a) [129], recommended as induction therapy in MPA/GPA, and mepolizumab (which reduces eosinophilic infiltration by blocking IL-5), recommended for induction and maintenance in EGPA [130]. Although there is a need for studies in children with AAV, avacopan has already been integrated with clinical guidelines that emphasise its glucocorticoid-sparing benefits [128,131].

Other therapeutic targets with similar mechanisms of action, including monoclonal antibodies against C5a (vilobelimab) currently tested in adult GPA/MPA and IL-5 blockade (benralizumab and depemokimab, both compared with mepolizumab in head-to-head clinical trials in adults with EGPA), are emerging [130].

More potent B cell depletion therapies, such as obinutuzumab, a type II anti-CD20 fully human monoclonal antibody with superior antibody-dependent cell-mediated cytotoxicity, are currently trialled in adults with AAV, while B cell-targeted combination therapy with belimumab (anti-BAFF) and rituximab has also been tested in adults with AAV [132]. There are trials exploring the efficacy of abatacept (CTLA4-Ig) in adult GPA [130], which, if successful, can lead to drug-repurposing in childhood AAV.

Based on experience and data derived from adult studies, we propose a potential management algorithm for GPA/MPA and EGPA in children, as shown in Figure 1.

Because of the scarcity of data derived from global childhood AAV cohorts, implementing this treatment algorithm may pose several limitations, from limited access to expertise in managing AAV in children, wide access to both established and novel therapies proposed here, in addition to challenges in accessing care or ensuring appropriate monitoring for many children and young people with AAV, and therefore, we advocate for multidisciplinary assessment of refractory cases and shared decision-making to support treatment choices. Virtual multidisciplinary meetings to provide guidance on the management of severe refractory cases as well as support access to compassionate treatment schemes may address some of these limitations.

Implementing such a treatment algorithm will not be possible without good-quality data regarding the safety and tolerability of emerging treatments in children and age/weight-appropriate dosing regimens, which can only be derived from high-quality clinical trials in global, ethnically and geographically diverse cohorts. Assessment of the long-term safety and efficacy of both established and novel therapies is also required.

## 13. Future Research

The last decade has marked significant progress in the treatment of AAV, with the advent of new targeted therapies involving various immune mechanisms relevant to AAV pathogenesis, from B cell depletion to complement, T cell, and IL-5 blockade. Even if the only targeted therapy currently recommended for use in AAV in children is rituximab, future inclusion of younger people in clinical trials will hopefully support the licensing of new effective therapies or re-purposing of existing ones, if proven effective, for childhood AAV as well. Recent research advances in the genetic characterisation of a large cohort of adult vasculitides highlighted a number of shared genetic variants with significance at the whole genome level, independently associated with more than two types of vasculitis, findings that have been exploited in terms of potential therapeutic target identification. The susceptibility loci identified suggest a further potential role of ustekizumab (IL-12/23 blockade) in ANCA -EGPA (associated with the *IL12B* susceptibility locus) cases or interferon I blockade in both ANCA-positive and negative EGPA (associated with the *IRF1* susceptibility locus) [58].

Future research should support large collaborative prospective studies to define outcomes of childhood AAV in the context of available therapies, as well as investment in paediatric interventional trials to test new therapies for AAV, including basket studies or using Bayesian design, as appropriate, considering the rarity of the condition and recruitment challenges. Additional investment should be directed towards a better understanding of the disease characteristics through the inclusion of populations that had very limited access to research historically [133]. This will enable us to understand how management strategies can be tailored based on individual characteristics, as well as to become aware of country-specific health system hurdles that we need to tackle as a priority to improve access to care for children with AAV worldwide. Initiatives such as the inclusion of specialists from all over the world in the CARRA and PReS vasculitis working parties have been welcomed by wider clinical and patient-expert communities.

Charities and patient advocacy groups play a critical role in identifying global unmet clinical needs for children and young people with AAV, while management recommendations should include specialists from across the globe to ensure that what expert health professionals propose as the gold standard for childhood AAV management is relevant and feasible to implement worldwide.

Ultimately, future research strategies should offer children and young people personalised management approaches to support sustained remission in AAV, minimisation of drug toxicity and severe flare risk, and preservation of good quality of life overall while providing suitable treatment options at different stages in their development and life (e.g., puberty, pregnancy) and appropriate comorbidity risk management to ensure adequate long-term outcomes.

## Figures and Tables

**Figure 1 ijms-25-13704-f001:**
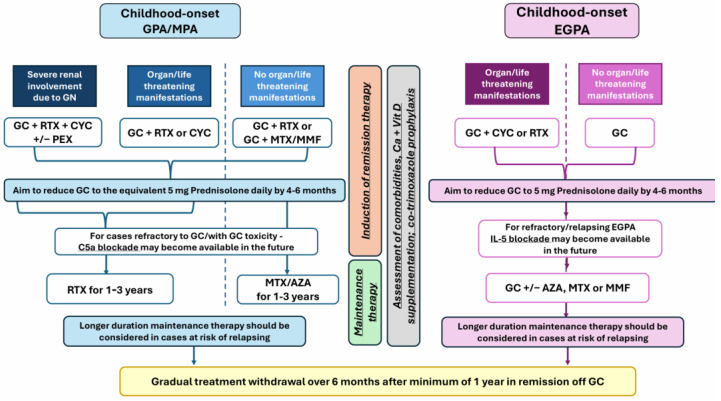
Legend: AZA—azathioprine; CYC—cyclophosphamide; EGPA—eosinophilic granulomatosis with polyangiitis; GC—glucocorticoids; GPA—granulomatosis with polyangiitis; MMF—mycophenolate mofetil; MPA—microscopic polyangiitis; MTX—methotrexate; PEX—plasma exchange; RTX—rituximab.

**Table 1 ijms-25-13704-t001:** Genetic polymorphisms associated with AAV phenotypes in children.

Genes/ Polymorphism	Geographical Area	Type of AAV	OR	Ref.
*HLA-DPB1*04:01*	European	GPA, Unclassified, MPA	3.5	Gibson et al., 2023 [54]
*ASB3 (Ankyrin Repeat And SOCS Box Containing 3)*	European	GPA, Unclassified, MPA	2.9	Gibson et al., 2023 [54]
*LINC02147 (long intergenic non-protein coding RNA 2147)*	European	GPA, Unclassified, MPA	5.7	Gibson et al., 2023 [54]
*ADNP (activity-dependent neuroprotector homeobox)*	European	GPA, Unclassified, MPA	6.6	Gibson et al., 2023 [54]

Novel mutations associated with AAV phenotypes in children.

**Table 2 ijms-25-13704-t002:** Novel mutations associated with AAV phenotypes in children.

Genes	Clinical Features	Paediatric Cases	References
*COPA (coatomer protein subunit alpha)*	Inflammatory lung disease, arthritis, renal disease	5-year-old girl with maternally inherited *COPA* variant c.679C>T (p.Arg227Cys): arthritis, AAV, progressive renal failure, minimal lung involvement	Zheng Y. et al., 2024 [98]
*ELANE (Elastase, Neutrophil Expressed)*	Severe congenital neutropenia, cyclic neutropenia, and rarely autoimmune manifestations	15-year-old boy with *ELANE* mutation c.103C>T, (p.R35X) (paternal) and *NF1* gene mutation c.5425C>T (p.R1809C): recurrent fever, rash, infection, lymphatic tuberculosis, SLE, lupus nephritis; 22-month-old girl with *ELANE* mutation c.452G>T (p.C151F): recurrent fever, infection, AIHA; 22-month-old boy with maternally inherited *ELANE* mutation c.242G>C (p.R81p): recurrent fever, infection, cyclic neutropenia, AAV	Zhang D. et al., 2023 [99]
*STING1 (Stimulator of Interferon Response CGAMP Interactor 1) or TMEM173 (transmembrane protein 173)*	SAVI (*STING*-associated vasculopathy with onset in infancy)—type I interferonopathy caused by dominant gain-of-function mutations in STING1. Early onset systemic inflammation with cutaneous vasculitis and interstitial lung disease	Hispanic girl with de novo pathogenic *STING1* gene mutation c.463G>A: SAVI-mimicking early onset ANCA vasculitis, symptoms beginning at 6 months, failure to thrive, recurrent fever, interstitial lung disease, pericarditis, pauci-immune ANCA (MPO-positive) vasculitis with diffuse necrotising and crescentic glomerulonephritis Three boys and three girls with *TMEM173* gene mutations c.461A>G (p.N154S), c.463G>A (p.V155M), and c.439G>C (p.V147L): symptoms within first 8 weeks, fever, telangiectatic +/− pustular +/− blistering rashes on cheeks, nose, fingers, toes, soles, manifestations of peripheral vascular inflammation (e.g., nodules on face, nose, ears; distal ulcers with infarcts), vascular and tissue damage (e.g., livedo reticularis, nail dystrophy/loss), pulmonary manifestations (e.g., paratracheal adenopathy; interstitial lung disease in five of the six patients); failure to thrive, recurrent infections 19-year-old male with pathogenic *STING1* mutation c.463G>A (p.V155M) presenting as adult-onset isolated renal and ocular ANCA-associated vasculitis. His father had the same mutation and presented with childhood-onset pulmonary fibrosis and renal failure attributed to ANCA-associated vasculitis.	Ochfeld E. et al., 2021 [100] Liu Y. et al., 2014 [101] Staels F. et al., 2020 [102]
APDS Activated phosphoinositide 3-kinase δ syndrome (APDS) 1 or 2 caused by gain/loss of function mutations in *PIK3CD* or *PIK3R1* genes, respectively	primary immunodeficiency causing recurrent infections (particularly sinopulmonary and herpes), immune dysregulation, lymphoproliferation, autoimmunity and malignancy	16-year-old female with *PIK3CD* gene mutation c.3061G > A (p. E1021K): cutaneous vasculitis, positive c-ANCA, elevated proteinase PR3-ANCA, recurrent sinopulmonary and herpesvirus infections, bronchiectasis, lymphadenopathy, increased transitional B cells, total and naïve T cell lymphopenia and dys-gammaglobulinemia 4-year-old boy and 7-year-old girl with *PIK3CD* gene mutation c.3061G>A (p. E1021K): lung infections, sinusitis, haematuria, positive ANCA, previously diagnosed as GPA 4-year-old Chinese boy with *PIK3CD* gene mutation c.1574A>G (E525G): sinusitis, otitis media, recurrent pneumonia, bronchiectasis, haematuria, lymphadenopathy, positive p-ANCA, initially diagnosed as AAV	Sood A.K. et al., 2023 [103] Lu M. et al., 2021 [104] Zhang X. et al., 2021 [105]

Legend: AIHA—autoimmune haemolytic anaemia; AAV—ANCA-associated vasculitis; SLE—systemic lupus erythematosus.

**Table 3 ijms-25-13704-t003:** Phonotypes raising the suspicion of a genetic cause for AAV.

Early onset in childhood
Family history of relatives with childhood-onset autoimmune conditions
Acutely unwell
Recurrent infections or association with lymphoproliferative conditions
Systemic inflammation associated with severe skin and internal organ manifestations
Laboratory tests suggestive of immunodeficiency and/or immune dysregulation: e.g., T, B, or NK cytopenias, neutropaenia, low complement levels or immunoglobulins
Overlapping clinical phenotypes
Refractory to treatment

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
