# Peer review of "Childhood-Onset ANCA-Associated Vasculitis: From Genetic Studies to Advances in Pathogenesis, Classification and Novel Therapeutic Approaches"

_ijms, 2024, doi:10.3390/ijms252413704_

Round 1
Reviewer 1 Report
Comments and Suggestions for Authors
To thank the authors for a review article on ANCA vasculitis in childhood.
A well-written article, complete and well separated by the main sections describing the disease.
It is always interesting to compare it with its presentation in adults, as it is a rare disease in childhood.
Only as a recommendation, reduce the section on genetic studies. It would be important to add a summary table explaining under what conditions a genetic study should be requested in ANCA vasculitisis in childhood.
Author Response
We would like to thank both reviewers for their useful comments.
We address them in their entirety and submitted a revised manuscript with tracked changes.
Reviewer 1 To thank the authors for a review article on ANCA vasculitis in childhood. A well-written article, complete and well separated by the main sections describing the disease. It is always interesting to compare it with its presentation in adults, as it is a rare disease in childhood.
Only as a recommendation, reduce the section on genetic studies. It would be important to add a summary table explaining under what conditions a genetic study should be requested in ANCA vasculitisis in childhood.
Thank you for these useful suggestions. The genetic studies paragraph has been amended with tracked changes (Pages 6-9). As suggested, we included a Table 3 with suggestions of clinical presentations that may raise the suspicion of a genetic cause for AAV.
Thank you for your time and consideration
Coziana Ciurtin
Reviewer 2 Report
Comments and Suggestions for Authors
How does using adult data affect the validity of conclusions about pediatric ANCA-associated vasculitis (AAV)? Are there any pediatric-specific data sources that could have been integrated?
Can the authors give more data or clarity to distinguish between correlation and causality in genetic markers related with pediatric AAV, such as HLA-DPB1*04:01 and LINC02147?
How do the genetic findings discussed in the paper translate into functional or mechanistic insights for pediatric AAV?
Given the limited availability of pediatric-specific data, how can the proposed therapeutic algorithms be effectively adapted for children in clinical settings, especially in regions with fewer resources?
Are there any potential risks or limitations to adopting treatment techniques from adult AAV to pediatric cases that the authors should address more explicitly?
How might differences in genetic and environmental factors between groups influence the findings and recommendations? Could including non-European cohorts increase the paper's applicability?
What are the hurdles to deploying innovative medicines like avacopan and mepolizumab in pediatric populations, and how may they be overcome?
What concrete efforts could researchers take to remedy the deficiencies in pediatric-specific AAV trials described in the paper?
How might international cooperation help overcome the low prevalence of pediatric AAV and improve research outcomes?
Author Response
We would like to thank you for your useful comments.
We addressed them in their entirety and submitted a revised manuscript with tracked changes.
Rewiewer 2
How does using adult data affect the validity of conclusions about pediatric ANCA-associated vasculitis (AAV)?
This is a relevant point, thank you.
The data we resented in relation to novel gene mutations, clinical presentation, serological characteristics and disease classification mainly pertain to studies in children. Many large GWAS study were performed in adults and contrasted with data from studies in children.
Less data however is available in relation to treatment options and there is a lack of good quality clinical trials in children. We have been guided in our treatment suggestions by evidence that classification criteria in adult AAV perform well in children and that there are commonalities in the disease pathogenesis, all suggesting that new treatments already proven effective in adults are very likely to work in children as well. These treatments are also suitable for young people with paediatric AAV when they reach adulthood. As these medications are not licensed in individuals younger than 18, there is no evidence to support these treatment options and therefore we included a discussion that’s why the proposal is rather aspirational.
We added relevant paragraphs (pages 2,6, 17) to explain these differences and their implications.
Are there any pediatric-specific data sources that could have been integrated?
We have integrated a few large cohorts and data sources in the paper as suggested, in addition to the ARChiVE/Pediatric Vasculitis Initiative (PedVas) registry, large French registry and EULAR/PReS/Ankara 2008 initiatives, under relevant paragraphs where we discussed clinical presentation, classification criteria, management strategies.
Additionally, we mentioned, the notable ongoing initiatives in place aiming underpinning large cohort studies supported by collaborative agreements and well-defined data collection protocols, such as the US/Canada Childhood Arthritis and Rheumatology Research Alliance (CARRA) AAV working group, the Paediatric Rheumatology International Trials Organisation (PRINTO) vasculitis database, the Pediatric Rheumatology European Society (EULAR/PReS) vasculitis working group currently involved in several multi-centre AAV studies, in addition to expert initiatives, such as the 2008 Ankara Consensus Conference, which have been all instrumental in advancing the knowledge related to childhood AVV – page 3 with tracked changes.
Can the authors give more data or clarity to distinguish between correlation and causality in genetic markers related with pediatric AAV, such as HLA-DPB1*04:01 and LINC02147?
This is an important point. We highlighted the lack of Mendelian randomization studies in childhood AVV, and difficulties in establishing the causative role of the genetic association described in childhood AAV, in addition to providing clinical guidance for genetic testing to identify potential causal novel gene mutations (Pages 6, 13, Table 3 with tracked changes)
How do the genetic findings discussed in the paper translate into functional or mechanistic insights for pediatric AAV?
All these aspects have been highlighted under each genetic association in the revised genetic paragraph- Pages 6-9 with tracked changes
Given the limited availability of pediatric-specific data, how can the proposed therapeutic algorithms be effectively adapted for children in clinical settings, especially in regions with fewer resources?
We agree that this very important to address and added the following paragraphs:
Although we recognize that the newly licensed treatment options for AAV have been evaluated in adult cohorts, there is currently enough evidence related to shared pathogenesis, adequate performance of classification criteria and unified established treatment approaches for AAV across the life span, to support our proposal for potential use of novel therapeutic approaches for childhood-onset AAV, as detailed in Figure 1. We also discussed below the limitations of extrapolating therapeutic strategies from adults to childhood AAV.
Are there any potential risks or limitations to adopting treatment techniques from adult AAV to pediatric cases that the authors should address more explicitly?
We added the following paragraphs:
Because of scarcity of data derived from global childhood AAV cohorts, implementing this treatment algorithm may pose several limitations from limited access to expertise in managing AAV in children, wide access to both established and novel therapies proposed here, in addition to challenges in accessing care or ensuring appropriate monitoring for many children and young people with AAV, and therefore, we advocate for multidisciplinary assessment of refractory cases and shared decision-making to support treatment choices. Virtual multidisciplinary meetings to provide guidance on management of severe, refractory cases, as well as support to access compassionate treatment schemes may address some of these limitations.
Implementing such a treatment algorithm will not be possible without good quality data regarding the safety and tolerability of emerging treatments in children, as well as age/weight appropriate dosing regimens, which can only be derived from high quality clinical trials in global, ethnically and geographically diverse cohorts. Assessment of long-term safety and efficacy of both established and novel these therapies is also required (Pages 17-18 with tracked changes)
How might differences in genetic and environmental factors between groups influence the findings and recommendations?
We have addressed all these aspects in the reviewed genetics/epigenetics/ environmental paragraphs Pages 6-9, with tracked changes.
Could including non-European cohorts increase the paper's applicability? What are the hurdles to deploying innovative medicines like avacopan and mepolizumab in pediatric populations, and how may they be overcome?
What concrete efforts could researchers take to remedy the deficiencies in pediatric-specific AAV trials described in the paper? How might international cooperation help overcome the low prevalence of pediatric AAV and improve research outcomes?
These are very important points, which we addressed in the reviewed paragraphs incorporated now into the manuscript.
Additional investment should be directed towards better understanding of the disease characteristics through inclusion of populations that had very limited access to research historically (155). This will enable us to understand how management strategies can be tailored based on individual characteristics, as well as become aware of country-specific health system hurdles we need to tackle as a priority to improve the access the care for children with AAV world-wide. Initiatives such as inclusion of specialists from all over the world in the CARRA and PReS vasculitis working parties have been welcomed by the wider clinical and patient-expert communities.
Charities and patient-advocacy groups have a critical role in identifying global unmet clinical needs for children and young people with AAV, while management recommendations should include specialists from across the globe to ensure that what expert health professionals propose as gold standard for childhood AAV management is relevant and feasible to implement world-wide (Page 18 with tracked changes).
Thank you for your time and consideration
Coziana Ciurtin
Round 2
Reviewer 2 Report
Comments and Suggestions for Authors
The paper can be accepted in its present form.